# Adaptive Support Ventilation Attenuates Ventilator Induced Lung Injury: Human and Animal Study

**DOI:** 10.3390/ijms20235848

**Published:** 2019-11-21

**Authors:** Yu-Ling Dai, Chin-Pyng Wu, Gee-Gwo Yang, Hung Chang, Chung-Kan Peng, Kun-Lun Huang

**Affiliations:** 1Graduate Institute of Medical Sciences, National Defense Medical Center, Taipei 114, Taiwan; oyday100@gmail.com; 2Division of Pulmonary and Critical Care Medicine, Tri-Service General Hospital, National Defense Medical Center, Taipei 114, Taiwan; 3Department of Critical Care Medicine, Taiwan Landseed Hospital, Tao-Yuan 32449, Taiwan; wucp@landseed.com.tw; 4Division of Chest Medicine, Buddhist Tzu Chi General Hospital, Hualien 970, Taiwan; ggyang@mail.tcu.edu.tw; 5Department of Physiology, National Defense Medical Center, Taipei 114, Taiwan; hung@ndmctsgh.edu.tw

**Keywords:** adaptive support ventilation, acute respiratory distress syndrome, ventilator-induced lung injury, surfactant depletion

## Abstract

Adaptive support ventilation (ASV) is a closed-loop ventilation, which can make automatic adjustments in tidal volume (V_T_) and respiratory rate based on the minimal work of breathing. The purpose of this research was to study whether ASV can provide a protective ventilation pattern to decrease the risk of ventilator-induced lung injury in patients of acute respiratory distress syndrome (ARDS). In the clinical study, 15 ARDS patients were randomly allocated to an ASV group or a pressure-control ventilation (PCV) group. There was no significant difference in the mortality rate and respiratory parameters between these two groups, suggesting the feasible use of ASV in ARDS. In animal experiments of 18 piglets, the ASV group had a lower alveolar strain compared with the volume-control ventilation (VCV) group. The ASV group exhibited less lung injury and greater alveolar fluid clearance compared with the VCV group. Tissue analysis showed lower expression of matrix metalloproteinase 9 and higher expression of claudin-4 and occludin in the ASV group than in the VCV group. In conclusion, the ASV mode is capable of providing ventilation pattern fitting into the lung-protecting strategy; this study suggests that ASV mode may effectively reduce the risk or severity of ventilator-associated lung injury in animal models.

## 1. Introduction

Acute respiratory distress syndrome (ARDS) has been studied for more than 50 years since its identification by Ashbaugh in 1967 [1]. ARDS can cause pulmonary endothelial and epithelial damage, which increases the permeability of the alveolar capillary barrier and leads to flooding of the alveoli with protein-rich pulmonary edema and activation of alveolar macrophages, which release proinflammatory chemokines and other cell factors. Neutrophil mobilization and activity increase, and hyaline membranes accumulate, which triggers simultaneous alveolar hemorrhage and edema, which in turn increase the pulmonary dead space and cause intrapulmonary shunting [2,3]. These events then impair lung compliance and oxygenation, and generate respiratory distress symptoms.

In 1998, animal model experiments performed by Dreyfuss et al. showed that high-volume ventilation can increase microvascular permeability and surfactant inactivation, which can then cause pulmonary edema, alveolar flooding, and leukocyte activation and infiltration, and finally result in distal lung tissue damage [4]. Slutsky’s and Chiumello et al. noted that the increase in transpulmonary pressures can cause lung stretching and that changes in transpulmonary pressure are inseparable from the volume changes in the lung [5,6]. The injuries generated by inappropriate tidal volume (V_T_) inflict damage on alveolar structure, and biological alterations proceed to cause further harm to the whole-body system. By contrast, low-stretch ventilation used with protective-ventilation strategies may limit further lung endothelial and epithelial damage, reduce the release of proinflammatory substances, and increase alveolar fluid clearance (AFC) through the activation of ion channels, such as the epithelial sodium channel and Na-K ATPase, which could in turn reduce edema and help in lung endothelial and epithelial recovery [7]. Another study has reported an association between mechanical ventilation and damage to the extracellular matrix (ECM) in alveolar epithelial cells [8]. These findings suggest that it may be possible to reduce the risk or severity of VILI by understanding the influence of ventilation mode on alveolar epithelial cells.

Adaptive support ventilation (ASV) is a type of closed-loop ventilation [9]. This mode follows the Otis rule [10] to achieve established ventilation goals while controlling the breathing work of the patient at a minimum. This mode can be switched between pressure control and pressure support according to the patient’s breathing drive, and can, thereby, provide full ventilation support [11,12,13]. Early studies of ASV emphasized mainly the shortening of the ventilator removal time [14,15,16,17]. Recent studies of the responses to ASV have focused on acute respiratory failure. Using human lung simulation and observations of patients, some researchers have noted the ability of ASV to maintain V_T_ when used with lung-protective strategies to treat acute lung injuries by providing a V_T_ in the range of 6–8 mL/kg [18,19,20,21,22], but the observations were limited to clinical physiological phenomena. Further study of the capability of automated ventilation to reduce the risk of VILI and the possible mechanisms is required. The purpose of this study was to further investigate how the ASV mode works when applied to ARDS patients. In order to study the mechanisms of ASV on human alveolar epithelial cells, we also performed experiments using an animal model of lavage-induced lung injury to investigate the mechanisms underlying the effects of ASV on alveolar epithelial cells and to investigate whether ASV could reduce the occurrence of VILI.

## 2. Results

### 2.1. Human Studies

Fifteen cases of ventilator-assisted treatment of patients diagnosed with ARDS were identified in the records from the ICUs of two medical centers in northern and eastern Taiwan. The patients were assigned to the ASV group or PCV group according to treatment. General information about these patients at the baseline is shown in Table 1. All of the patients in the two groups were diagnosed with pneumonia and were treated in the chest medicine department. The distribution of age, sex, and acute physiology and chronic health evaluation II (APACHE II) scores did not differ significantly between the two groups. The ideal body weight was 61.9 ± 6.23 kg for the ASV group and 57.2 ± 7.33 kg for the PCV group. Before the treatment, the basic respiratory physiological parameters and ventilator settings, hemodynamic parameters, and use of sedatives, muscle relaxants, and inotropic drugs did not differ significantly. The main outcomes, such as the mortality and ventilator weaning, also showed no statistical difference between groups.

The changes in the lung mechanics parameters and breathing patterns of the two groups shown in Figure 1. The inspiratory pressure in all patients was <30 cmH_2_O. In addition, PEEP, compliance, respiratory rate, tidal volume, and minute ventilation have changes in baseline, 24 h and 72 h, and there are no statistical differences.

After being included in the case for 24 h, whether the two ventilation modes were able to maintain a V_T_ < 8 mL/kg intended to provide lung protection and the associations with mortality rates (Appendix A). In the PCV group, three patients (50%) had a V_T_ >8 mL/kg and all died. In the ASV group, only one patient (12%) had a Vt > 8 mL/kg and survived. Due to the small number of cases in each group, the overall survival rate is not statistically different.

### 2.2. Animal Experiments

Eighteen 10- to 12-week-old piglets were included in the animal experiment according to the experimental procedure (Appendix A). Each animal was assigned to one of three groups (*n* = 6 per group): control, VCV, and ASV groups. Their average weights were 25.3 ± 3.88, 25.2 ± 4.49, and 23.3 ± 3.78 kg, respectively.

#### 2.2.1. Physiological Parameters

The three groups all maintained hemodynamic stable (as shown in Appendix A) throughout the experiments. The VCV group had a higher heart rate (>120 beats/min) and a lower mean arterial blood pressure (<80 mmHg) after the lavage-induced lung injury (*p* < 0.05 compared with the other two groups). The ELWI values measured with the PiCCO after the lung injury inductions were maintained between 12.2 and 14.2 mL/kg in the control group and between 17.5 and 21.8 mL/kg for the VCV and ASV groups; the differences were significant (*p* < 0.05) between each of the two experimental groups and the control group, but not between the two experimental groups. The mean cardiac index was between 3.71 and 4.96 L/min m^−2^ for the three groups, but did not differ significantly between any groups.

The minute ventilation was set at 5.2–6.0 L/min for the three groups (Table 2) using the mechanical ventilation settings throughout the experiments. The ASV group maintained ventilation with a low tidal volume (VT) of 6.3–6.6 mL/kg and a higher respiratory rate (RR) of 35 and 36 beats/min after induction lung injury. A fixed V_T_ of 8 mL/kg was maintained in the VCV group by using a higher peak airway pressure of 26.2–28.2 cmH_2_O, which was significantly higher than that in the control and ASV groups.

EIT measurements were performed using one animal per group to demonstrate non-invasive ventilation images (Appendix A, Appendix A). In EIT image, the blue silhouette shows the intra-thoracic gas content. The magnitude of impedance change is color-coded and ranges from dark blue to white, which respectively indicates low to high impedance changes. A high tidal impedance image means high ventilation area. The changes in the EIT images were consistent with the changes in physiological ventilation parameters in this study. A consistent tidal impedance image was maintained at every time point in the control group. After lung injury, the high ventilation area was lower in the ASV group than those in the control group. The nadir of decrease in high ventilation area was 3 h after lung injury.

In the experiments, the ventilation parameters were recorded for over 30 min, and the coefficients of variance (CVs) of the parameters were calculated as shown in Table 3. The baseline and post-lavage CVs were consistent in the three groups, but minute ventilation, V_T_, respiratory rate and inspiratory flow rate 3 h after injury were significantly higher in the ASV group than in the two other groups.

The changes in gas exchange variables at different time points are shown in Figure 2. As shown in Figure 2a, pH decreased from 7.45 to 7.33 after lung injury in the VCV group, and was significantly lower in the VCV group than in the control group. The pH value in the ASV group was between those of the other two groups and did not differ significantly from either group. As shown in Figure 2b, after lung injury, the *P*/*F* ratio were <100 in the two experimental groups. However, the *P*/*F* ratio recovered to 135 ± 19 at 1hour and to 168 ± 21 at 3 h after lavage in ASV group but remained <100 in the VCV group 1–3 h after lavage. Figure 2c shows the dead space ratio after lung injury in the three groups. This ratio was maintained between 25.3 ± 2.8 and 26.8 ± 2.2 in the VCV group and between 16.7 ± 2.8 and 11.5 ± 2.3 in the ASV group; the *p* < 0.05 compared with the VCV group. The pulmonary shunt fractions did not differ significantly between the groups (Figure 2d).

#### 2.2.2. Indicators of Lung Injury

Before the end of the experiments, bronchopulmonary lavage was performed and BALF was acquired for the measurement of indicators of lung injury. The total protein concentrations in the BALF from the three groups are shown in Figure 3a. Compared with the control group (274.8 ± 56.4 μg/mL), the concentration was higher in the VCV group (3679.9 ± 1273.1 μg/mL) and ASV group (1008.2 ± 180.8 μg/mL) after lung injury; *p* < 0.05 in the ASV group compared with the VCV group.

The cytokine concentrations in BALF were measured (Appendix A). Concentrations of IL-6 (Appendix A) and TNF-α (Appendix A) in BALF were significantly higher in the two experimental groups than in the control group, but the concentrations of IL-1β and IL-8 did not differ significantly between any groups. Cytokine concentrations in the blood did not differ between groups at any time (Appendix A).

FITC–dextran–albumin was injected through a bronchoscope, and BALF was collected after 1 and 30 min. The changes in the concentration of the collected FITC–dextran–albumin solution were determined and used to calculate AFC. As shown in Figure 3b, the AFC of the ASV group was 12.9% ± 1.47% before the end of the experiment; this value was similar to that in the control group. By contrast, the AFC was −0.71% ± 2.81% in the VCV group because of the inability to discharge the fluid effectively within the alveoli; this finding indicates severely impaired alveolar epithelial cell function.

Lung tissue was obtained, and the wet and dry weights were measured to evaluate the degree of edema. The lung W/D ratio was 8.45 ± 0.6 in the VCV group (Figure 3c), which was significantly higher than the ratio of 6.3 ± 0.2 in the control group. The ratio was intermediate in the ASV group (7.88 ± 0.60) and did not differ significantly from the other two groups.

#### 2.2.3. Epithelial Barrier Functions

Gelatin zymography was performed on eight BALF samples from the control, VCV, and ASV groups (*n* = 3 in each group). Gelatinolytic activity bands were detected at about 63 kDa for matrix metalloproteinase 2 (MMP-2) (pro-MMP-2) and at 80 and 90 kDa for MMP-9 (pro-MMP-9 and high-molecular-weight forms, respectively) (Figure 4a,b). The peak areas of the high-molecular-weight bands of MMP-9 differed significantly between groups. The peak area was significantly higher in the VCV group (3,314,333 ± 169,778) than in the control group (1,012,430 ± 271,885) and slightly higher than the ASV group (2,226,180 ± 908,459). This increased production of MMP-9 suggests excessive lung inflammation and tissue destruction in VCV group.

Because AFC is a critical aspect of recovery after lung injury, we used western blotting to measure claudin-4 levels in the three groups (*n* = 3 in each group). Claudin-4 abundance was significantly lower in the VCV group than in the other two groups (Figure 4c,d). We also measured the level of occluding, a tight junction protein that plays a critical role in maintaining the integrity of lung epithelial barrier. Occludin abundance was also significantly lower in the VCV group than in the other two groups (Figure 4e,f). These findings suggest that ventilation-induced lung injury and alveolar edema are related to the degradation of claudin-4 and occludin.

## 3. Discussion

To our knowledge, this is the first study to compare the use of the automated ASV ventilation mode in human clinical studies and in an animal model. We examined the effects of the use of the ASV mode in patients with ARDS. Our results showed that the ASV mode was able to control the V_T_ in the range of 6–8 mL/kg IBW in most ARDS patients, suggesting highly feasible to apply this mode as part of the lung-protective ventilation strategy. Regardless of whether the PCV or ASV mode was affected by the patient’s own respiratory drive, only a few patients had a V_T_ >8 mL/kg IBW 24 h after initiation of treatment (Appendix A) and three of them in PCV group all died. These results further demonstrate the importance of the ability of automated breathing modes to maintain a low V_T_ in ARDS treatment.

This study is also the first to focus on the influence of the ventilation mode on alveolar epithelial barrier functions. We applied ASV in a pig model of lung injury. The ASV group exhibited a rapid shallow breathing pattern, which was consistent with those described in the ASV clinical studies by Arnal et al. [23,24]. We also obtained ventilation images during single experiments using EIT, and these images showed a decrease in V_T_ 1–3 h after lung injury in the ASV group (Appendix A). This finding is similar to that obtained from the images 1–4 h after lung injury in pig experiments following the ARDSNet protocol reported by Pomprapa’s research group [25]. Compared with the VCV group, the ASV group had a better *P*/*F* ratio and *V_D_*/*V_T_* ratio (Figure 2). These findings suggest that in the pig model the ASV mode could generate a breathing pattern, which is similar to the lung-protective ventilation strategy used in humans. A study by Chiumello et al. found that a high V_T_ is more likely to increase the stress and strain in the lungs of ARDS patients, which may cause further lung damage [26]. Therefore, when applying mechanical ventilation, a lower V_T_ should be used to keep the lung strain to a minimum and the ASV mode may act as a lung-protective strategy by producing a minimal strain to the lung.

In our animal experiments, the CVs of the breathing patterns in the ASV group increased, in contrast to the results in the VSV group. Although the animals were completely anesthetized and their respiratory drive did not influence the experimental process, we found significant increases in the CVs for the number of breaths, V_T_, and inspiratory flow rate 1–3 h after injury in the ASV group. This may have occurred because of a decrease in alveolar cyclic stretch caused by the automated control mechanisms of the ASV mode. Our results are similar to those reported by Samary et al. who found improved lung damage by using variable ventilation, which increased V_T_ variability, in an animal model [27]. Spieth et al. also found that adding a variable VT mode reduced the extent of interstitial pulmonary edema and lung epithelial cell damage, in their animal model [28]. Recently, Rentzsch and other researchers observed a lower synthesis of proinflammatory cytokines after variable stretch to the lipopolysaccharide-stimulated alveolar epithelial cells, in contrast to nonvariable stretch [29]. In our study, cytokines were detected in BALF after the experiment ended. The concentrations of all BALF cytokines were higher after lung injury in the two experimental groups than in the control group (Appendix A). However, the serum cytokine concentrations did not differ significantly at any time points (Appendix A). These results are compatible to the experimental findings of Muellenbach et al. [30], suggesting that the lung inflammation in the experimental animals originated from the direct mechanical damage, but not from a systemic inflammatory response. Our results also showed that the total protein content in BALF was lower (Figure 3a) and the AFC was higher in the ASV group than in the VCV group (Figure 3b). These results suggest that ASV may attenuate lung injury by reducing mechanical damage and local lung inflammation.

This study is one of the few to assess the effects of mechanical ventilation mode on alveolar epithelial cell barrier functions in an animal model in which we attempted to identify the mechanisms responsible for alveolar epithelial cell damage. It has been well documented that the stretching of alveolar epithelial cells caused damage to the tight junction structures [31]. MMPs are endopeptidases that are capable of degrading extracellular matrix proteins, among whichMMP-9 acts to degrade the alveolar ECM, and triggers changes in cell structures in lung diseases [32]. Lanchou et al. have detected an increase in MMP2 and MMP9 in the early stages of ARDS in BALF obtained from ARDS patients [33]. Pirrone et al. reported changes in MMP-2 and MMP-9 in pig experiments in different lung injury models and reported that mechanical injury induced by a ventilator could increase MMP-2 and MMP-9 in lung tissue [34]. MMP-9 level has also been reported to be high in BALF in ARDS animal models [32]. Pirrone et al. reported that mechanical injury induced by a ventilator could increase MMP-2 and MMP-9 in pig lung tissue [34]. In our study, the concentration of MMP-9 in BALF (Figure 4a,b) increased after lung injury in both experimental groups, although the concentration was higher in the VCV group than in the ASV group. In contrast to the elevation of MMPs, the presence of claudin-4 is associated with less ECM degradation. Rokkam et al. demonstrated in human lungs that high claudin-4 protein expression in lung tissues of donors was associated with a high AFC% and tissues with a lower lung injury score had a higher expression of claudin-4 [35]. The same phenomenon was observed in this study showing that claudin-4 expression (Figure 4c,d) and AFC% (Figure 3) were much higher in the ASV group than in the VCV group. These findings suggest that mechanical damage occurred in the VCV group and that intermediate values in the ASV group may reflect less mechanical damage.

In summary, this study demonstrated a high feasibility to apply ASV mode as part of the lung-protective ventilation strategy in ARDS patients. Moreover, our animal study showed that an increase in the breathing pattern variance with the ASV ventilation mode is associated with reduced mechanical damage from alveolar cyclic stretch, decreased ECM degradation, and improved barrier functions of alveolar epithelial cells.

This study has several limitations. First, only 15 ARDS patients could be recruited for the human studies, and the group allocation by the randomization center produced different numbers in the two groups, which may have resulted in a failure to find significant difference because of an insufficient number of patients. Second, the breathing patterns selected in the animal experiments differed from those in the human study. Future studies of the respiratory CVs could focus on pressure-control modes to understand the variability in respiratory patterns in different modes. Finally, the EIT measurements to image ventilation changes were obtained for only one experimental animal, and we could not perform a thorough analysis of the changes in ventilation induced by the different modes.

## 4. Materials and Methods

Additional details are provided in online Appendix A.

### 4.1. Human Studies

#### 4.1.1. Ethics Statement

This study was approved through research process reviews by the research ethics committees of two medical centers in the Taiwan area (approval numbers: TSGHIRB 097–05–175 14 January 2009 and Tzu-Chi General Hospital IRB098–35, 26 June 2009). All participants or their family members signed informed consent forms for the study.

#### 4.1.2. Study Design

This study was a prospective randomized clinical research trial. The data for 15 patients diagnosed with ARDS who received ventilator-assisted treatment in the intensive care unit (ICU) of two medical centers in Taipei and eastern Taiwan from June 2009 to November 2011 were collected. The ARDS diagnosis was based on the criteria of the American European Consensus Conference on ARDS as follows: ratio of the arterial partial pressure of oxygen to the fraction of inspired oxygen (FiO_2_) (*P*/*F*) < 300 mmHg, chest X-ray showing bilateral infiltrates, and no evidence of left heart failure (pulmonary capillary wedge pressure (PCWP) < 18 mmHg) when available. The V_T_ was targeted to align with the lung-protecting strategies as closely as possible for all patients. The patients were randomly allocated to groups using sealed envelopes containing allocation information prepared beforehand by the randomization center within 48 h of the start of ventilator use. The patients were randomly allocated a pressure control ventilation (PCV) group (*n* = 6) or an ASV group (*n* = 9). The experimental allocation process and ventilation setting are shown in Appendix A.

#### 4.1.3. Measurements

The main variables evaluated for the ASV and PCV modes included ventilator detachment rate, ventilator use duration, length of stay in the ICU and hospital, and mortality rate. The secondary variables were related to the breathing patterns and physiological parameters, and included variations in breathing patterns, such as the oxygenation index, partial oxygen and carbon dioxide pressures in arterial blood, number of breaths, V_T_, and minute ventilation.

### 4.2. Animal Experiments

The animal experimental procedures in this study were approved by the Animal Care and Use Committee of the National Defense Medical Center (approval number: IACUC-12–139 12 July 2012; IACUC-14–162, 24 June 2014).

#### 4.2.1. Animal Preparation and Cannulation

Ten to 12-week-old male pigs weighing 20–30 kg were fasted for 12 h, but supplied with water ad libitum, before anesthesia. The pigs were premedicated with tiletamine–zolazepam (Zoletil forte veterinary use; Virbac, Carros, France) at a dose of 5 mg/kg (2.5 mg/kg of zolazepam and 2.5 mg/kg of tilazamine) and atropine (0.1 mg/kg) given intramuscularly before the anesthesia. Anesthesia and neuromuscular blockade were induced with pentobarbital sodium (10–18 mg/kg h^−1^), fentanyl (10–26 mg/kg·h^−1^), and cisatracurium (0.1 mg/kg·h^−1^). A PiCCO catheter (PULSION Medical Systems, Munich, Germany) was inserted into the right femoral artery and was used for the measurement of cardiac parameters and pulmonary edema. A Swan–Ganz catheter (Edwards Lifesciences, Irvine, CA, USA) was wedged into a pulmonary arteriole through the central venous catheter for mean pulmonary arterial pressure and PCWP measurements [36].

#### 4.2.2. Induction of ARDS by Surfactant Deactivation

After the pig was stabilized, the lungs were lavaged in the supine and alternating lateral decubitus positions using a 5% solution of Tween 20 in saline at 3–5 mL/kg instilled into the right, dependent lung beyond the tracheal bifurcation. Following lavage, the endotracheal tube was reconnected to the ventilator for three breaths, and the lungs were then inflated from 0 to 30 cmH_2_O and held for 10 s for each breath. This procedure was repeated three times to evenly distribute the lavage fluid. The pig was then rotated to the left lateral decubitus position and the Tween lavage procedure was repeated in the left lung [37].

#### 4.2.3. Experimental Protocol

Eighteen pigs in total were monitored for arterial blood pressure and hemodynamic parameters using A-lines and were subjected to mechanical ventilation (G-5 ventilator, Hamilton Medical, Rhäzuns, Switzerland) during the tracheostomy. The pigs were allocated to one of three mechanical ventilation mode groups: control group, volume-control ventilation (VCV) group, and ASV group (*n* = 6 for each group). For all three groups, the baseline volume control mode V_T_ was 10 mL/kg, the positive end-expiratory pressure (PEEP) was 5 cmH_2_O, and the FiO_2_ was 50%. After injury induction in the VCV group, the V_T_ was set at 8 mL/kg and the number of breaths was set to maintain the end-tidal CO_2_ pressure (EtCO_2_) at 35 to 45 mmHg. For the ASV group, the ASV% was adjusted via the ASV mode to provide minute ventilation similar to that in the VCV group and was fine-tuned to maintain EtCO_2_ at 35 to 45 mmHg.

During the experimental procedure, the recording time points were the baseline, immediately after lavage ARDS, and 1, 2, and 3 h after ARDS. At every time point, the physiological breathing parameters, blood flow parameters, and data from gas analysis of arterial and venous blood were recorded, and blood was collected for measurement of cytokine concentrations. Before the experiment was terminated, pulmonary fluid clearance was measured, and the lung lavage fluid was collected using bronchoscopy. Lung tissue was collected for the analyses of the lung tissue wet/dry (*W*/*D*) ratio and other associated proteins through open-chest surgery before the animal was sacrificed. The experimental allocation process is shown in Appendix A.

#### 4.2.4. Physiological Measurements

The extravascular lung water index (ELWI) was measured using the PiCCO system by applying the thermodilution technique as follows: 10 mL of 4 °C normal saline was input three times via the central venous catheter and the mean cardiac output index was calculated after the three measurements. Hemodynamic parameters—including heart rate, mean arterial pressure, and mean pulmonary arterial pressures—were recorded. Physiological breathing parameters, such as airway pressure, V_T_, and flow rate were measured breath-by-breath through a proximal pneumotachograph (single-use flow sensor, PN 279331, Hamilton Medical, Bonaduz, Switzerland), which was linear between −120 and 120 L/min with ± 5% error of measure at the experimental time points. The records were input into a computer using Datalogger software (version 3.27, Hamilton Medical, Bonaduz, Switzerland) via an RS232 cable for storage. The recorded data included ventilator settings (such as predicted body weight, percentage mechanical ventilation, PEEP, and F_I_O_2_), breathing patterns (V_T_, total and spontaneous respiratory rate, inspiratory time, expiratory time), and breathing mechanical parameters (such as lung compliance, expiratory resistance, expiratory time constant, and intrinsic PEEP). The breathing parameters were recorded for more than 1 min at the baseline, at the time of ARDS, and continuously for more than 40 min after ARDS induced.

#### 4.2.5. Alveolar Fluid Clearance

Sixty milligrams of fluorescein isothiocyanate (FITC)–dextran–albumin (Sigma-Aldrich, St. Louis, MO, USA) was dissolved in and diluted with 30 mL of Ringer’s lactate solution. Before sacrifice, bronchoscopy was performed. The bronchoscope was inserted into the right lung lobe and fixed in position, and the diluted solution containing 1–1.5 mL/kg FITC–dextran–albumin was sent into the right lung lobe via a number 60 PE tube within the bronchoscope. One minute later, the PE tube was changed for a clean one, and the albumin lavage fluid within the lung was withdrawn using an empty syringe and obtained for analysis of the basal alveolar fluid clearance (C_initial_). Thirty minutes later, the PE tube was changed, the albumin lavage fluid was withdrawn, and the lavage fluid was obtained for analysis of the alveolar fluid clearance (C_final_). The alveolar FITC–dextran–albumin lavage fluids from both drawings were centrifuged, and the fluorescence intensity was measured using a microplate reader (HT. 202905, BioTek Instruments, Winooski, VT, USA). The C_initial_ and C_final_ values were used to calculate the AFC ratio as follows: AFC (% decrease) = ((C_initial_ − C_final_)/C_initial_) × 100 [38,39].

#### 4.2.6. Ratio of Lung *W*/*D* Weight

The lung *W*/*D* weight ratio was used as an indicator of lung water accumulation after the induction of ARDS. To quantify the total amount of lung water, the animals were dissected under deep sevoflurane anesthesia, and the lung weight was measured immediately after its excision (wet weight). The lung tissue was then dried in an oven at 60 °C for 5 days and weighed again for dry weight values. The *W*/D weight ratio was calculated by dividing the wet weight by the dry weight.

#### 4.2.7. Bronchoalveolar Lavage Fluid (BALF)

The pigs were sacrificed 3 h after the induction of ARDS, and the lungs were lavaged twice with saline. The BALF was then distributed into microcentrifuge tubes, to which an anti-proteinase mixture had been added as follows: 100 mM phenyl methyl sulfonyl fluoride in isopropanol was first added at a concentration of 10 μL/mL BALF, followed by 500 mM of ethylenediaminetetraacetic acid in double-distilled water per 10 μL/mL BALF. The BALF sample tubes were then flash-frozen in liquid N_2_ and stored at −20 °C.

#### 4.2.8. Total Protein Analysis

Total protein concentration in BALF, an indicator of epithelial and cell membrane integrity, was quantified using the modified Lowry Method Protein Assay Kit (Thermo Fisher Scientific, Rockford, lL, USA). In brief, this involves a modified trichloroacetic acid protein precipitation reaction followed by a modified Lowry method for colorimetric concentration determination. Sample absorbance was recorded at a wavelength of 600 nm on a spectrophotometer (BioTek Instruments, Winooski, VT, USA). For the total protein analysis, BALF samples were diluted 1:1 with dH_2_O.

#### 4.2.9. Measurement of Proinflammatory and Oxidative Stress Cytokine Concentrations

The concentrations of the proinflammatory and oxidative stress cytokines interleukin 1β (IL-1 β), IL-6, IL-8, and tumor necrosis factor α (TNF-α) in BALF were measured using sandwich enzyme-linked immunosorbent assay (ELISA) kits purchased from eBioscience. The assay diluent and tetramethyl benzoate substrate (TBS) reagent sets for IL-6, IL-8, and TNF-α were purchased from BD Biosciences. For IL-1β, the assay diluent used was a solution of TBS with 3% bovine serum albumin (BSA) and 0.05% Tween 20 (pH 7.3). All standard curves for the ELISAs were generated as indicated in the respective kit instructions. The samples were diluted appropriately for each assay, and their final absorbance was scanned at 450 nm using a microplate reader.

#### 4.2.10. Gelatin Zymography

Gelatin zymography was performed using copolymerizing gels and 7.5% sodium dodecyl sulfate-polyacrylamide gel electrophoresis (SDS-PAGE) with gelatin (0.1%) (Sigma-Aldrich, St. Louis, MO, USA. For each sample, 20 μg of total BALF protein was loaded. Electrophoresis was performed using a Mini Protean 3 Minigel slab apparatus (Bio-Rad) at a constant voltage of 120 V and was terminated when the dye reached the bottom of the gel. Following electrophoresis, the gels were washed in a renaturation buffer (2.5% Triton X-100 in 50 mm Tris–HCl (pH 7.5)) for 1 h in an orbital shaker. The zymograms were incubated for 18 h at 37 °C in an incubation buffer (0.15 m NaCl, 10 mm CaCl_2_, and 0.02% NaN_3_ in 50 mm Tris–HCl (pH 7.5)). The gels were then stained with Coomassie blue and destained with 7% methanol and 5% acetic acid. Areas of enzymatic activity appeared as clear bands over a dark background [40].

#### 4.2.11. Western Blotting for Claudin-4 and Occludin

Total protein in samples from the control, VCV, and ASV groups was subjected to SDS-PAGE (40 μg samples). The proteins were transferred to a nitrocellulose membrane (Millipore, Burlington, MA USA), followed by blocking with 5% BSA (Sigma-Aldrich, St. Louis, MO, USA) for 1 h at room temperature (RT) with gentle shaking. The membranes were then incubated with antibodies to claudin-4 (1:100 dilution), occludin (1:100 dilution; both from Invitrogen, Carlsbad, CA, USA), and β-actin (1:10,000 dilution; Sigma-Aldrich, St. Louis, MO, USA) overnight at 4 °C with mild shaking, followed by incubation of membranes with horseradish peroxidase-conjugated anti-rabbit (1:20,000 dilution) and anti-mouse (1:50,000 dilution; both from Jackson Laboratory, Bar Harbor, ME, USA), respectively, for 1 h at RT with gentle shaking. The blots were washed three times with Tris-buffered saline. The enhanced chemiluminescence assay (Thermo Fisher Scientific, Rockford, lL, USA) was used to develop the membrane, and autoradiography was performed using high-performance chemiluminescence film according to the manufacturer’s guidelines.

#### 4.2.12. Statistical Analysis

All statistical analyses were performed using IBM SPSS Statistics for Windows (ver. 19.0) (IBM Corp., Armonk, NY, USA). The data are shown as mean ± standard deviation (SD). *p* values <0.05 were considered to be significant. Repeated-measures ANOVA was used to identify differences between time points. One-way ANOVA was used to identify differences between groups, and this was followed by the Student–Newman–Keuls method to locate the differences.

## 5. Conclusions

The ASV ventilation mode was capable of providing breathing patterns closely aligned to the lung-protective strategies. In the animal model, the CVs were higher for the breathing patterns of the ASV ventilation mode than those of the VCV ventilation mode. This suggests that providing ventilation with lower airway pressure and V_T_ is associated with a better *P*/*F* ratio and alveolar epithelial barrier functions. This study suggests that ASV mode may effectively reduce the risk or severity of ventilator-associated lung injury in animal models.

## Figures and Tables

**Figure 1 ijms-20-05848-f001:**
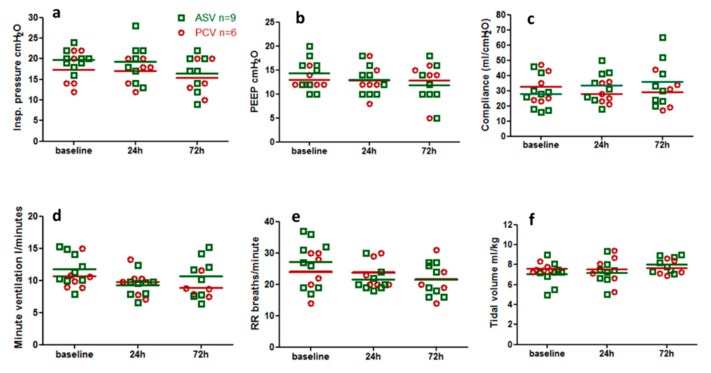
(**a–c**) Changes in respiratory mechanics and (**d–f**) changes in minute ventilation and respiratory patterns at different time points in the two patient groups. ASV, adaptive-support ventilation; PCV, pressure control ventilation; baseline, the time of enrollment in the study; 24 h, 24 h after enrollment in the study; 72 h, 72 h after enrollment in the study; PEEP, positive end-expiratory pressure; RR, respiratory rate.

**Figure 2 ijms-20-05848-f002:**
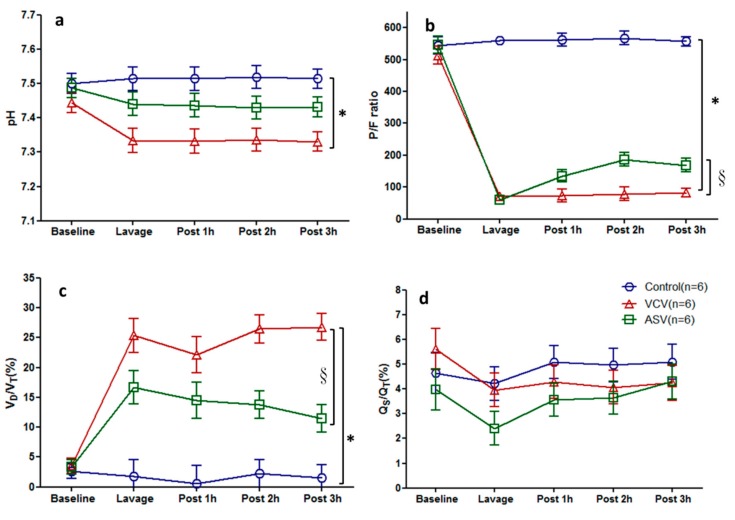
Time course of gas exchange in the three experimental groups (*n* = 6 in each group). (**a**) pH, (**b**) *P*/*F* ratio, (**c**) dead space (*V_D_*/*V_T_* ratio =(PaCO_2_-PetCO_2_)/PaCO_2_), (**d**) pulmonary shunt fractions (Qs/Qt = (CcO_2_ − CaO_2_)/CcO_2_ − CvO_2_)). The data are presented as the mean ± SD. ^*^
*p* < 0.05 compared with the control group; ^§^
*p* < 0.05 compared with the VCV group.

**Figure 3 ijms-20-05848-f003:**
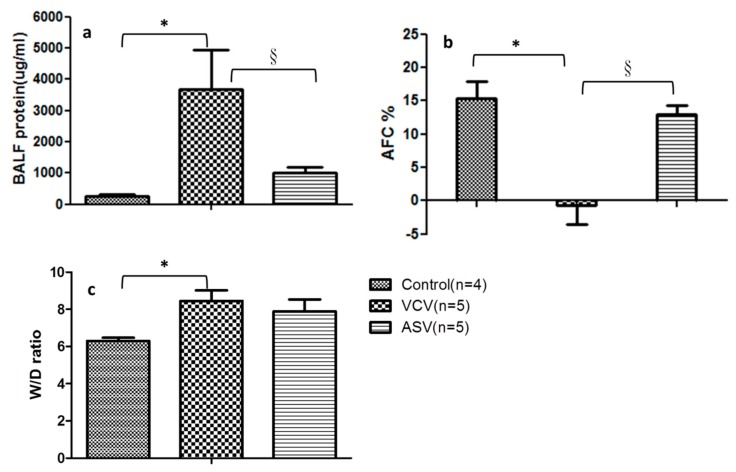
Indicators of lung injury in the experimental groups. (**a**) BALF total protein content, (**b**) alveolar fluid clearance (AFC) rates, and (**c**) the wet-to-dry ratio of lung tissue. ^*^
*p* < 0.05 compared with the control group; ^§^
*p* < 0.05 compared with the VCV group.

**Figure 4 ijms-20-05848-f004:**
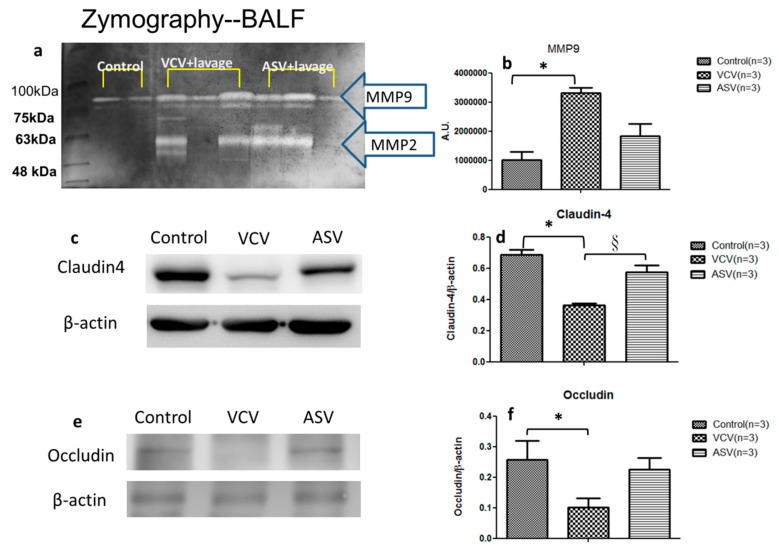
Gelatinase activity in BALF and expression of claudin-4 and occludin in the three experimental groups. (**a**) Bands corresponding to MMP-2 (63 kDa) and MMP-9 (80 kDa) are highlighted. (**b**) Quantitative data for MMP-9 are presented as mean ± SD. (**c**) Western blots of tissue samples show the difference between groups in claudin-4 abundance. (**d**) Quantitative data for claudin-4 in each group were normalized to β-actin. (**e**) Western blots of tissue samples show difference between groups in occludin abundance. (**f**) Quantitative data for occludin for each group were normalized using β-actin. ^*^
*p* < 0.05 compared with the control group; ^§^
*p* < 0.05 compared with the VCV group.

**Table 1 ijms-20-05848-t001:** Baseline characteristics and outcome in human study

Variable	ASV Group (*n* = 9)	PCV Group (*n* = 6)	*p* Value
Age, yr, mean ±SD	62.0 ± 18.5	62.5 ± 30.3	0.776
ApacheⅡscores, mean ±SD	17.0 ± 5.44	21 ± 8.46	0.524
Gender, male/female (%)	7/2(77.8/22.2)	4/2(66.7/33.3)	1
**Baseline Condition**			
IBW, kg, mean ±SD	61.9 ± 6.23	57.2 ± 7.33	0.145
PaO_2_/FiO_2_, mean ±SD	153.6 ± 53.6	156.7 ± 32.8	1
Static compliance, ml/cmH_2_O, mean ±SD	28.0 ± 10.6	32.7 ± 10.5	0.529
Pressure level, cmH_2_O, mean ±SD	19.8 ± 2.79	17.3 ± 4.5	0.456
PEEP level, cmH_2_O, mean ±SD	14.8 ± 3.15	12.4 ± 0.9	0.066
Minute volume, l/min, mean ±SD	11.8 ± 2.52	10.7 ± 2.26	0.388
Tidal volume/IBW, ml/kg, mean ±SD	7.06 ± 1.2	7.61 ± 0.38	0.272
Total respiratory rate, breaths/min, mean ±SD	27.1 ± 7.5	24.0 ± 6.5	0.529
Mean BP, mmHg, mean ±SD	90.4 ± 16.9	84.5 ± 28.8	0.607
Sedation +/−, (%)	8/1(88.9/11.1)	6/0(100/0)	1
Muscle relaxants +/−, (%)	4/5(44.4/55.6)	2/4(33.3/66.7)	1
Inotropic drug +/−, (%)	1/8(11.1/88.9)	3/3(50/50)	0.235
**Main Outcome**			
Wean success / fail *n*. (%)	5/4(55.6/44.4)	2/4(33.3/66.7)	0.608
Ventilator-free days, median (IQR)	7(4–9)	8(4.5–17)	0.473
Days in ICU, median (IQR)	15(12.5–16.5)	13(5.5–21.5)	0.314
Days of hospitalization, median (IQR)	29(19–55)	15(11–43.5)	0.174
Alive/death *n*. (%)	4/5(44.4/55.6)	2/4(33.3/66.7)	0.608

Values are expressed as mean (SD) unless indicated otherwise. ASV, adaptive support ventilation; PCV, pressure-control ventilation; APACHE II, acute physiology and chronic health evaluation II; IBW, ideal body weight; *P*/*F*, ratio of arterial partial oxygen pressure to inspiratory oxygen concentration; PEEP, positive end-expiratory pressure; mean BP, mean arterial blood pressure; IQR, interquartile range; ICU, intensive care unit. Sedative agents included midazolam and propofol. Muscle relaxants included Nimbex and Pavalon. Inotropic agents included dopamine, noradrenaline, and vasopressin. Data were compared using the Mann–Whitney *U* test or χ^2^ test.

**Table 2 ijms-20-05848-t002:** Respiratory parameters and lung mechanics in animal experiments.

Parameter	Group	Baseline	Lavage	Post 1 h	Post 2 h	Post 3 h
V_E_, L/min						
	Control (*n* = 6)	5.2 ± 0.86	5.1 ± 0.86	5.1 ± 0.81	5.1 ± 0.82	4.9 ± 0.69
	VCV (*n* = 6)	5.7 ± 0.62	5.7 ± 0.57	6.0 ± 0.37	5.9 ± 0.34	5.9 ± 0.33
	ASV (*n* = 6)	5.5 ± 1.1	5.3 ± 1.0	5.5 ± 0.53	5.5 ± 0.62	5.5 ± 0.62
V_T_, mL/kg						
	Control	10.0 ± 0.26	9.9 ± 0.31	10.0 ± 0.31	9.9 ± 0.35	10.0 ± 0.36
	VCV	10.1 ± 0.58	9.3 ± 0.88	8.0 ± 0.21^&*^	8.0 ± 0.28^&*^	7.9 ± 0.34^&*^
	ASV	9.9 ± 0.42	9.9 ± 0.05	6.4 ± 0.93^&*^^§^	6.6 ± 1.03^&*^^§^	6.3 ± 1.03^&*^^§^
f, per minute						
	Control	20.4 ± 2.8	20.5 ± 3.2	20.2 ± 3.2	20.2 ± 3.1	19.7 ± 2.3
	VCV	23.1 ± 5.6	25.5 ± 7.1	30.8 ± 6.1^&*^	30.0 ± 4.9^&*^	30.0 ± 4.8^&*^
	ASV	23.9 ± 3.9	22.8 ± 3. 1	35.7 ± 3.9^&*^	35.5 ± 4.5^&*^	36.1 ± 4.9^&*^
P_Peak_, cmH_2_O						
	Control	18.9 ± 3.4	18.0 ± 4.5	17.6 ± 4.3	18.1 ± 4.6	18.5 ± 4.7
	VCV	20.1 ± 2.2	28.3 ± 5.2^&*^	26.2 ± 3.6^&*^	27.0 ± 4.4^&*^	28.2 ± 5.1^&*^
	ASV	18.5 ± 2.8	27.2 ± 4.3^*^	19.5 ± 1.9^§^	20.0 ± 1.5^§^	21.7 ± 1.0^§^
C_st_, mL/cmH_2_O						
	Control	23.6 ± 5.3	26.1 ± 6.6	26.4 ± 6.3	25.6 ± 5.8	25.1 ± 5.7
	VCV	21.8 ± 5.4	13.9 ± 1.6^&*^	12.1 ± 1.5^&*^	11.3 ± 1.4^&*^	10.8 ± 1.8^&*^
	ASV	22.5 ± 4.9	15.1 ± 4.2^&*^	12.7 ± 4.3^&*^	12.2 ± 3.7^&*^	10.7 ± 2.3^&*^
R_rs_, cmH_2_O/L/s						
	Control	7.2 ± 1.2	7.7 ± 1.9	7.9 ± 1.9	8.1 ± 2.4	7.9 ± 1.9
	VCV	9.6 ± 3.3	9.1 ± 2.5	6.3 ± 2.2	6.7 ± 3.4	6.8 ± 4.3
	ASV	7.4 ± 1.4	10.8 ± 2.6	9.8 ± 2.1	8.9 ± 2.5	9.3 ± 2.9

Values are expressed as the mean and SD. V_E_, minute ventilation; ASV, adaptive support ventilation; VCV, volume control ventilation; V_T_, tidal volume; f, respiratory frequency; P_peak_, peak airway pressure; C_st_, static lung compliance; R_rs_, resistance of the respiratory system. ^&^
*p <* 0.05 compared with the baseline; * *p <* 0.05 compared with the control group; ^§^
*p <* 0.05 compared with the VCV group. Differences between and within groups were tested using general linear model statistics and were adjusted for repeated measurements.

**Table 3 ijms-20-05848-t003:** Coefficients of variance in different group breathing patterns.

Parameter	Group	Baseline	Lavage	Post 1 h	Post 2 h	Post 3 h
V_E_, %						
	Control (*n* = 6)	0.29 ± 0.16	0.3 ± 0.21	0.5 ± 0.32	0.65 ± 0.69	0.41 ± 0.26
	VCV (*n* = 6)	0.76 ± 0.65	1.0 ± 0.41	2.5 ± 3.21	0.73 ± 0.82	0.53 ± 0.28
	ASV (*n* = 6)	0.78 ± 0.86	1.9 ± 1.9	2.6 ± 0.96	2.25 ± 1.06 ^*^^,^^§^	1.6 ± 0.71 ^*^^,^^§^
V_T_, %						
	Control	0.43 ± 0.14	0.49 ± 0.21	0.59 ± 0.2	0.66 ± 0.21	0.67 ± 0.24
	VCV	0.69 ± 0.37	0.67 ± 0.31	0.76 ± 0.29	0.61 ± 0.22	0.68 ± 28
	ASV	0.61 ± 0.32	1.49 ± 1.29	3.7 ± 3.35 ^*^^,^^§^	2.8 ± 0.96 ^*^^,^^§^	1.9 ± 0.22 ^*^^,^^§^
f, %						
	Control	0 ± 0	0 ± 0	0.14 ± 0.22	0.26 ± 0.64	0.15 ± 0.36
	VCV	0 ± 0	0 ± 0	0.29 ± 0.42	0.07 ± 0.17	0.14 ± 0.18
	ASV	0 ± 0	0 ± 0	3.6 ± 2.97 ^*^^,^^§^	2.23 ± 1.12 ^*^^,^^§^	1.56 ± 0.8 ^*^^,^^§^
Insp. flow, %						
	Control	0 ± 0	0 ± 0	0 ± 0	0 ± 0	0 ± 0
	VCV	0 ± 0	0 ± 0	0.18 ± 0.44	0 ± 0	0 ± 0
	ASV	0 ± 0	0 ± 0	3.6 ± 0.71 ^*^^,^^§^	4.98 ± 2.7 ^*^^,^^§^	3.94 ± 2.2 ^*^^,^^§^

V_E_, minute ventilation; ASV, adaptive support ventilation; VCV, volume-control ventilation; V_T_, tidal volume; f, respiratory frequency. ^*^
*p* < 0.05 compared with the control group; ^§^
*p* < 0.05 compared with the VCV group. CV was calculated as (SD/mean) × 100.

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
