# Peer review of "Adaptive Support Ventilation Attenuates Ventilator Induced Lung Injury: Human and Animal Study"

_ijms, 2019, doi:10.3390/ijms20235848_

Round 1
Reviewer 1 Report
A strong and a very interesting article .The scientific soundness will be very strong. The novelty of the article is very high. It is an important article for the doctors and researchers.
The article related to adaptive support ventilation in human and animal study is an excelent scientific paper with a high originality data.
The English is very well.
The statistical data analysis is well done. The introduction cover an important background informations.
The discussion section is very good.
Author Response
Response to Reviewer 1 Comments
Point 1: A strong and a very interesting article .The scientific soundness will be very strong. The novelty of the article is very high. It is an important article for the doctors and researchers.
The article related to adaptive support ventilation in human and animal study is an excellent scientific paper with a high originality data.
The English is very well.
The statistical data analysis is well done. The introduction cover an important background informations.
The discussion section is very good.
Response 1: Thank you for your comments.
Reviewer 2 Report
Abstract
Insert the rationale to apply ASV to decrease the VILI
Which are the main determinants of VILI
How ASV and PCV were setted
Inclusion criteria early or late ARDS
Just 15 patients enrolled thus this is not necessary to reach an adequate power to detect any difference in the mortality
Lower alveolar strain stress (please clarify)
Stress or strain ??
The conclusions are not based on the present findings
Main TEXT
Insert line 52 slutsky studies Chiumello et al
Insert the recent review paper on esophageal pressure and transpulmonary pressure published in the american journal
Is a problem of ventilation AVS vs PCV or a settings related problems
How was PEEP fixed ??
There is not any method paragraph, which is very important
There is not any explanation of the measurements performed and also the time
Please insert a power analysis to justify the small sample size population, 15 patients
Consecutive or not ???
For example several data are not well explained: BAL, intrapulmonary shunt, dead space, EIT
Results
To long and to many data, try to reduce the length and the number of data
Baseline characteristic insert the PEEP level
Try to concentrate the major data in few tables
Animal data
Tidal volume was very high around 10 ml/Kg
Page 6 line 159 3 hours after injury
Please insert a table of the EIT data
Discussion
Why difference in compliance of respiratory system
I suggest to focus only on animal data, human data are a very small size
I am wondering on the total number of measurements that you performed, such as EIT, shunt, BAL
The discussion is too long and difficult to follow
Please delete ALI in favour of ARDS
Author Response
Response to Reviewer 2 Comments
Point 1: Abstract:
Insert the rationale to apply ASV to decrease the VILI Which are the main determinants of VILI How ASV and PCV were setted Inclusion criteria early or late ARDS Just 15 patients enrolled thus this is not necessary to reach an adequate power to detect any difference in the mortality Lower alveolar strain stress (please clarify), Stress or strain ?? The conclusions are not based on the present findings.Response 1: Abstract
Adaptive support ventilation (ASV) is a volume-targeted pressure support/pressure control mode that can automatically adjust tidal volume (VT) and respiratory rate (RR) based on the minimum work of breathing principle of Otis, may automatically select a smaller tidal volume to achieve a lung protection strategy.According to the relationship of Intrathoracic Pressures and Lung Stretching described in Reference 5 of this paper, it can be known that the main factor causing VILI is lung stress and strain.
Because Abstract cannot be described in detail by word count, the ventilation setting protocol will be described supplementary materials (Fig S3).
Human inclusion criteria: 1.Acute onset <48 hrs, 2.PaO2/FiO2 <300, 3.Bilateral infiltrates consistent with pulmonary edema, 4.No clinical evidence of left heart failure
Human studies have been closed for two years (June 2009 and September 2011) due to difficulties patient recruitment, so it has not reached statistical differences. This article only discusses clinical phenomena.
Thanks for reminding, corrected. (Lower alveolar strain)
Thanks for reminding, corrected.
Point 2: Main TEXT
Insert line 52 slutsky studies Chiumello et al. Insert the recent review paper on esophageal pressure and transpulmonary pressure published in the american journal Is a problem of ventilation AVS vs PCV or a settings related problems How was PEEP fixed ?? There is not any method paragraph, which is very important There is not any explanation of the measurements performed and also the time Please insert a power analysis to justify the small sample size population, 15 patients, Consecutive or not ??? For example several data are not well explained: BAL, intrapulmonary shunt, dead space, EITResponse 2: Main TEXT
Thanks for reminding, corrected and insert the new references.Thanks for reminding, insert the human studies ventilation setting protocol to the supplementary materials (Fig S3).
In the human studies the PEEP level were not fixed; In the Animal experiments the PEEP level were fixed.
Thanks for reminding, limited by article length the method paragraph put into supplementary materials. (Fig. S1-S3)
Thanks for reminding, the description in supplementary information.
Human studies have been closed for two years due to difficulties patient recruitment, so it has not reached statistical differences. This article only discusses clinical phenomena.
Thanks for reminding, the relevant parameters have been explained
Point 3: Results
Too long and too many data, try to reduce the length and the number of data Baseline characteristic insert the PEEP level Try to concentrate the major data in few tablesAnimal data
Tidal volume was very high around 10 ml/Kg Page 6 line 159 3 hours after injury Please insert a table of the EIT dataResponse 3: Results
1,3. Thanks for your suggestion, in order to reduce the length of the table1 combined the baseline characteristics and outcome into one table, but still try to reduce one column (Section)
Thanks for reminding, corrected
Animal data
Animal experiment was designed according to the previous study, (References-36, 37) Vt 10 ml / kg for the control group and baseline condition.Thanks for reminding, corrected.
Thanks for reminding, because EIT instruments are available for measurement in the late stages of the experiment; only one animal per group is measured as a demonstration of non-invasive ventilation images. To reduce the length of the article, move the EIT image and data table to supplementary materials (Fig.S6, TableS1).
Point 4: Discussion
Why difference in compliance of respiratory system I suggest to focus only on animal data, human data are a very small size I am wondering on the total number of measurements that you performed, such as EIT, shunt, BAL The discussion is too long and difficult to follow Please delete ALI in favour of ARDS
Response 4:
In the animal experiment, VCV and ASV groups after the Tween lavage induced ARDS, the Surfactant was washout, and the compliance of respiratory system was significantly lower than that of Control group and Baseline condition.Thanks for your suggestion; the human data does not have statistical power, so this article only in discussion section1 discusses clinical phenomena.
The number of specimens collected by each experimental project is different, indicated in the figure or table. Shunt-each groups n=6(Fig3); BALF cytokine- control (N=5), VSV (N=6), and ASV (N=6) groups (FigS6); EIT image-each groups n=1 (Fig.S6, TableS1)
Thanks for your suggestion; we try to streamline the discussion section.
Thanks for reminding, because of the respect for the original, some literature references still use ALI.
Round 2
Reviewer 2 Report
No further comments
Author Response
Thank you for your review!